# MindShot: Multi-Shot Video Reconstruction from fMRI with LLM Decoding

## Abstract

Reconstructing dynamic videos from fMRI is important for understanding visual cognition and enabling vivid brain-computer interfaces. However, current methods are critically limited to single-shot clips with video-level alignment and reconstruction, failing to address the multi-shot nature of real-world experiences. To bridge this gap, we propose MindShot, a novel shot-level framework that effectively reconstructs multi-shot videos from fMRI via a divide-and-decode strategy. Specifically, our framework consists of three stages: (1) Shot Decomposition: We first identify shot boundaries within fMRI, then decompose the mixed signals into distinct, shot-specific segments. This explicit segmentation serves as the foundation for accurate semantics decoding. (2) Keyframe Decoding: Each segment is decoded into a textual description representing the keyframe of its corresponding shot. (3) Video Reconstruction: The final video is generated from these keyframe captions, effectively mitigating noise from fMRI redundancy. Addressing the critical lack of real data for multi-shot reconstruction, we introduce a large-scale synthetic dataset generated via a novel data augmentation strategy that randomizes scene duration ratios. Experimental results demonstrate our framework outperforms state-of-the-art methods in both single-shot and multi-shot reconstruction fidelity. Crucially, ablation studies confirm the necessity and generalizability of our Shot Boundary Predictor (SBP), where explicit shot-level decomposition significantly improves decoded caption CLIP similarity by 71.8%, and the SBP yields consistent performance gains when integrated into other state-of-the-art architectures. Moreover, our synthetic data makes the model generalizable to diverse data and has strong zero-shot transferability that effectively bridges the domain gap between synthetic and real fMRI signals. This work establishes a new paradigm for multi-shot fMRI reconstruction, enabling accurate recovery of complex visual narratives through explicit decomposition and semantic prompting.

## 1 Introduction

Functional magnetic resonance imaging (fMRI) is a powerful, non-invasive tool for studying the human brain, particularly the visual system, through indirect measurement of neural activity(Horikawa & Kamitani, 2017). Reconstructing dynamic visual sequences from fMRI data is critical not only for advancing our understanding of dynamic visual perception and cognition, but also for developing next-generation brain-computer interfaces (BCIs) capable of more vivid and dynamic "mind-reading" applications (Wen et al., 2018; Fang et al., 2020; 2023). However, existing video reconstruction research mainly focuses on short-duration, single-shot videos (depicting a single, continuous scene or event) (Sun et al., 2025; Chen et al., 2023; Li et al., 2024; Lu et al., 2025), ignoring the multi-shot visual experiences that characterize real-world cognition, such as watching films or recalling episodic memories.

Reconstructing multi-shot video presents substantial challenges beyond single-shot reconstruction, especially for existing video-level paradigm. Whether aggregating signals over longer sequences or decoding short, fixed-length clips, these approaches attempt to reconstruct entire video clips from corresponding brain activity segments. This paradigm, however, critically neglects that natural videos are often composed of multiple semantic events. Consequently, when the fMRI signals correspond to multiple scenes, the video-level approach leads to temporal mixing of semantically distinct neural patterns. This not only confines current methods to effectively handling only single-

shot videos but also introduces significant ambiguity and noise into the reconstructions, making it challenging to disentangle and accurately reconstruct the separate visual events (Figure 1).

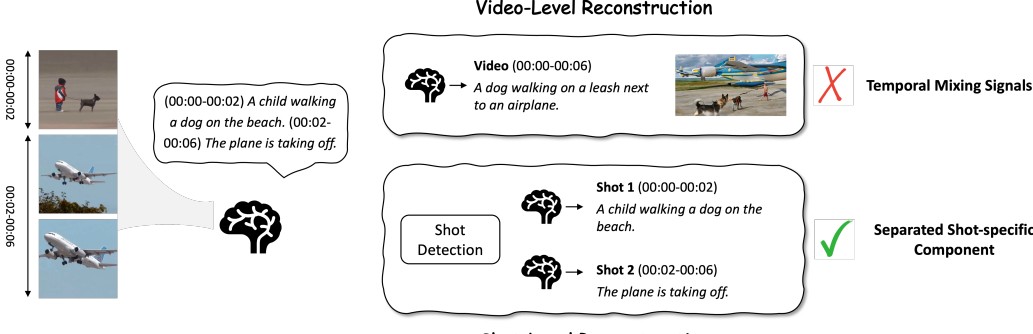

Figure 1: Illustration of the key innovation of our proposed shot-level paradigm. The conventional video-level framework decodes from temporally mixed fMRI signals, leading to semantically entangled results. In contrast, our shot-level paradigm explicitly decomposes the signals into shot-specific components before decoding, enabling clean and coherent reconstruction of each shot.

To address this limitation, we propose MindShot, a novel divide-and-decode framework for multi-shot video reconstruction. Instead of aggregating the entire fMRI data, we introduce a learnable shot boundary predictor that segments fMRI into shot-specific components corresponding to individual shots, enabling explicit and independent reconstruction of each shot. To address the limitations imposed by the fMRI-video temporal resolution mismatch, we propose to decode keyframe captions from fMRI data to achieve semantically precise reconstruction. For each segmented shot-specific fMRI, we decode a textual caption describing the keyframe using Large Language Models (LLMs). This leverages the observation that humans primarily encode salient events at a high semantic level (Doerig et al., 2025), making the LLM-decoded caption robust against the inherent temporal blurring leading by hemodynamic response function (HRF). The decoded caption then provides a precise semantic prompt for the subsequent video generation stage. To overcome data scarcity, we develop novel synthesis strategies to construct a large-scale multi-shot fMRI-video dataset. Leveraging existing publicly available fMRI-video datasets, including the benchmark CC2017 (Wen et al., 2018) and the dataset by (Chen et al., 2023), we synthesize 20k sample pairs for each dataset with randomized scene ratios, enhancing the robustness of our proposed model. Our contributions in this work can be summarized as follows:

- We introduce a new shot-level paradigm for fMRI-video reconstruction by establishing the shot as the fundamental unit of decoding. This shot-level paradigm enables the explicit reconstruction of complex, multi-shot videos for the first time.

- We design a learnable shot boundary predictor (SBP) that automatically segments fMRI time series into shot-specific components, effectively mitigating the temporal mixing problem without manual intervention. Experiments show that SBP is model-agnostic and yields consistent performance gains when integrated into existing methods, demonstrating broad applicability.

- We propose a novel LLM-based Keyframe Decoding module that decodes high-fidelity semantic prompt from shot-specific fMRI segment, effectively solving the challenge of the inherent noise present in fMRI signals. Ablation studies demonstrate that our LLM-based text-only decoding yields better reconstruction metrics than the fMRI-only alternative.

- We develop novel synthesis strategies to create large-scale multi-shot training data from existing datasets, facilitating model development for multi-shot video reconstruction. Ablation studies confirm that the high fidelity of this synthetic data, demonstrating strong zero-shot transferability and effective data augmentation for enhanced temporal robustness.

## 2 RELATED WORK

### 2.1 FMRI-TO-IMAGE RECONSTRUCTION

Benefiting from large-scale datasets like the Natural Scenes Dataset (NSD) (Scotti et al., 2023), generative vision models conditioned on fMRI signals have demonstrated unprecedented performance in reconstructing static images from brain responses. Existing research primarily focuses on enhancing reconstruction fidelity through improved semantic alignment, such as contrastive learning techniques that align fMRI embeddings with image or text representations (Xia et al., 2024), or by incorporating low-level image features to preserve visual detail consistency (Wang et al., 2024). Additional efforts have developed subject-unified methods to address cross-subject alignment and model generalization (Scotti et al., 2024). Despite significant progress, reconstructing dynamic video sequences presents substantially greater challenges than static images.

### 2.2 FMRI-TO-VIDEO RECONSTRUCTION

As a pioneering work of fMRI-to-video reconstruction, MindVideo (Chen et al., 2023) achieves notable fidelity by aligning fMRI features to CLIP (Radford et al., 2021) space for latent diffusion model prompting. Subsequent studies enhance temporal modeling in fMRI encoders (Sun et al., 2025) or explore cross-subject alignment via fMRI projection (Li et al., 2024). Crucially, most of existing methods are confined to single-shot scenarios, neglecting the multi-shot dynamics inherent in real-world cognition. While NeuroClips (Gong et al., 2024) generates multiple shots by fusing semantically similar keyframes, it relies on post-hoc processing rather than intrinsic fMRI signal decomposition, failing to optimize encoders for disentangling mixed shot information within fMRI windows. Moreover, contrastive alignment in video reconstruction may be challenging due to the temporal resolution mismatch between fMRI and video. In contrast to prior work, we propose to explore the multi-shot video reconstruction by shot-specific fMRI segmentation and keyframe caption decoding for semantically precise reconstruction, circumventing contrastive alignment constraints.

## 3 METHOD

Our method can be divided into three main stages, as shown in Figure 2. In the first stage, the shot boundary predictor partitions fMRI into shot-specific components. Each segmented fMRI is then decoded to shot-specific keyframe caption via direct interaction with a LLM. These captions serve as precise semantic prompts input to a text-to-video diffusion model for final video synthesis.

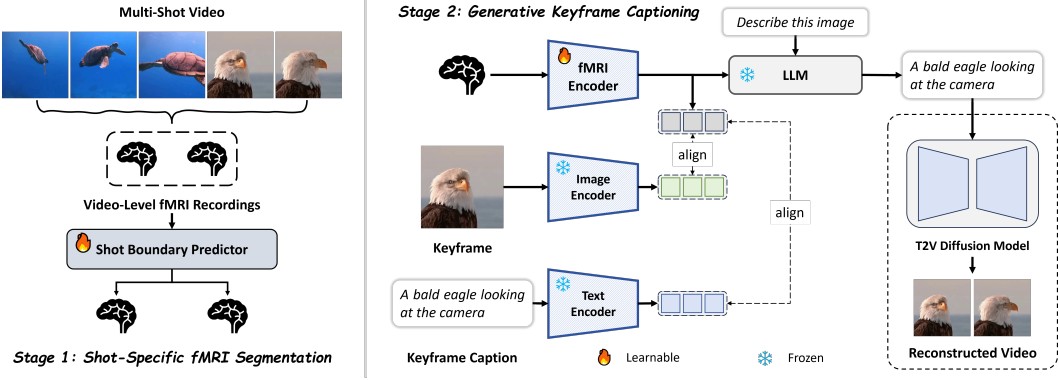

Figure 2: Overview of our proposed method, consisting of three main stages: Shot-Specific fMRI Segmentation, Generative Keyframe Captioning, and Shot-Centric Video Reconstruction.

### 3.1 DATASET SYNTHESIS

To address the scarcity of multi-shot fMRI data, we construct synthetic datasets based on CC2017 (Wen et al., 2018) and fMRI-WebVid (Li et al., 2024).

**Synthesis from fMRI-WebVid** Using 2,000 single-shot training clips and 400 test clips from fMRI-WebVid (each 4s video with 5 fMRI scans), we synthesize multi-shot samples by randomly concatenating distinct clips into 4s sequences. We enforce diversity by varying shot duration ratios (e.g., 2:3, 5:0). Middle-frame keyframes are extracted and captioned via BLIP-2 (Li et al., 2023), yielding 20,000 training and 1,000 test samples with aligned annotations.

**Synthesis from CC2017** From 1,440 training and 400 test clips in CC2017 (each 6s video with 3 fMRI scans), we synthesize two-shot videos by concatenating distinct clips while maintaining 6-second duration. The number of synthesized fMRI scans is set to 4, considering about the duration of each shot. We first employ SceneSeg (Rao et al., 2020) to decompose single-shot segments. Two distinct shots are then concatenated with randomized duration ratios (e.g., 3:1) to form 6s videos corresponding to 4 fMRI scans. Adopting the same keyframe and captioning protocol, we generate 20,000 training and 1,000 test pairs.

## 3.2 SHOT-SPECIFIC FMRI SEGMENTATION

A primary challenge in multi-shot video reconstruction is the temporal mixing of neural signals across different shots. Achieving semantically precise reconstruction thus requires decomposing video-level fMRI into shot-specific components. The most intuitive approach is to detect shot boundaries that occur across successive fMRI volumes (TRs), transforming the problem into sequence boundary prediction for subsequent fMRI separation and aggregation. Therefore, we propose to detect boundaries by introducing a shot boundary predictor. It is important to note that due to the temporal integration inherent in fMRI signal acquisition within a single TR, resolving scene transitions that occur entirely within one TR presents a fundamentally different and more challenging problem. Specifically, our shot boundary predictor aims to detect macroscopic transitions between neural representations rather than instantaneous pulses. This is feasible given event segmentation principles (Zacks et al., 2007; Baldassano et al., 2017) and evidence for fine-grained fMRI temporal decoding (Wittkuhn & Schuck, 2021).

Leveraging the fMRI encoder from (Li et al., 2024), we get fMRI embeddings $emb_f \in \mathbb{R}^{M \times c}$, where $M$ is the number of fMRI scans, and $c = 1024$ represents the embedding dimension. Theses embeddings are then processed by our proposed shot boundary predictor, which comprises a two-layer bidirectional LSTM (Bi-LSTM) to model bidirectional temporal dependencies in fMRI signals, and a linear layer generating boundary probabilities.

Formally, given fMRI embeddings $emb_f \in \mathbb{R}^{M \times c}$, the boundary probabilities are computed as $H = \text{Bi-LSTM}(emb_f)$, where $H \in \mathbb{R}^{M \times d}$ are hidden states ($d = 512$), $P = WH + b \in [p_1, p_2, \ldots, p_{M-1}]$ denotes boundary probabilities and $p_i$ represents the boundary probability of a boundary between fMRI scans $i$ and $i + 1$.

The model is optimized via binary cross-entropy loss:

$$\mathcal{L}_{sbp} = -\frac{1}{M-1} \sum_{i=1}^{M-1} [y_i \log p_i + (1 - y_i) \log(1 - p_i)] \tag{1}$$

where $y_i \in \{0, 1\}$ indicates ground-truth boundaries, and the true number of shots $N$ satisfies $N = 1 + \sum_{i=1}^{M-1} y_i$. At inference, the binarized boundaries $o_i$ is set to 1 if $p_i$ exceeds a threshold $\tau$, and 0 otherwise. In this work, the threshold $\tau$ is set to 0.5. Using the binarized boundaries $o_i$, we partition the fMRI sequence into $\tilde{N}$ segments, where $\tilde{N} = 1 + \sum_{i=1}^{M-1} o_i$ is the predicted number of shots. The shot-specific embeddings are then aggregated as $emb_f^s \in \mathbb{R}^{\tilde{N} \times c}$.

## 3.3 GENERATIVE KEYFRAME CAPTIONING

Beyond temporal signal mixing, the fMRI-video temporal resolution mismatch makes direct reconstruction via contrastive alignment challenging. However, human cognition encodes experiences through semantic abstractions of key events rather than continuous visual streams. We therefore reformulate the task as keyframe-centered semantic reconstruction, where decoding keyframe captions bypasses strict temporal alignment requirements. Specifically, we learn to generate keyframe captions directly from shot-specific fMRI signals using an LLM, and the keyframe captions are then used for final video synthesis.

Using ground-truth shot boundaries during training, we obtain shot-specific fMRI embeddings $emb_f^s \in \mathbb{R}^{N \times c}$ and concatenate them with an instruction prompt and input into a frozen LLM. To align these continuous signals with the frozen LLM, each fMRI token undergoes a projection module to match the dimensionality of the LLM's token embedding space. These projected vectors are then prepended as a continuous prefix to the text prompt embeddings. The resulting input follows a structured dialogue format: System: [system message]. User: $<$ instruction $> <$ fMRI embedding $>$. Assistant: $<$ answer $>$. The tag $<$ instruction $>$ denotes natural language query, while $<$ fMRI embedding $>$ is a placeholder for fMRI embedding. Crucially, the LLM attends to these projected vectors as dense contextual information analogous to prefix-tuning, rather than interpreting them as discrete textual tokens. This allows the model to leverage its multimodal understanding to generate captions based on the fMRI-visual context. The objective for optimizing this decoding process is to minimize the text modeling loss $\mathcal{L}_{caption}$, which evaluates the ability of LLM to generate target captions from fMRI embeddings. This loss is formally defined as the negative log-likelihood of the target captions given context:

$$\mathcal{L}_{caption} = -\sum_{k=1}^{T} \log P_\theta \left( t_k | t_{<k}, I; emb_f^s \right) \tag{2}$$

where $T$ is the length of target text, $t_k$ is the $k$-th token, $t_{<k}$ represents the preceding tokens, $I$ is the input prompt ('Describe this image $<$ image $>$' in this work), and $P_\theta$ is the token probability distribution parameterized by LLM weights $\theta$.

We empirically found that introducing contrastive alignment and noise prediction during training can improve the final results. Given {keyframe, keyframe caption} pairs, CLIP loss is calculated for fMRI-keyframe and fMRI-caption pairs, $L_{align}$ calculates the mean CLIP loss between the fMRI embedding and both the keyframe visual embedding and the text embedding. The MSE loss for noise prediction also guides the video generated conditioned on fMRI embedding. During this training phase, the U-Net of the video diffusion model is frozen. $L_{mse}$ measures the difference between the predicted noise and the ground-truth noise. The overall training loss combines all components:

$$\mathcal{L} = \mathcal{L}_{sbp} + \lambda_1 \mathcal{L}_{caption} + \lambda_2 \mathcal{L}_{align} + \lambda_3 \mathcal{L}_{mse} \tag{3}$$

where $\lambda_1$, $\lambda_2$, and $\lambda_3$ are learnable parameters for automatic optimization. Only the fMRI encoder and shot boundary predictor are trained while other modules remain frozen.

### 3.4 SHOT-CENTRIC VIDEO RECONSTRUCTION

Following the generation of keyframe captions, we reconstruct the final video by generating each shot individually and concatenating them according to their original duration. The video generation is conditioned only on the textual captions using a frozen text-to-video diffusion model.

## 4 EXPERIMENT AND RESULTS

### 4.1 EXPERIMENTAL SETTING

**Dataset.** We evaluated our method on both synthesized and original datasets, including CC2017 (Chen et al., 2023) and fMRI-WebVid (Li et al., 2024). Both utilize 3T fMRI scanners, but differ in TR and clip length. For synthesized datasets, we balanced samples across different duration ratios to provide diversified data. CC2017-Syn used fMRI ratios of [(0,4), (1,3), (2,2), (3,1)] for 4 synthesized fMRI scans, while fMRI-WebVid-Syn used ratios of [(0,5), (2,3), (3,2)]. Synthesized training data originated only from original training data with no test overlap.

**Evaluation Metrics** For video reconstruction, we utilized N-way top-K accuracy for semantic evaluation and SSIM for pixel-level assessment. Shot-specific fMRI segmentation employed segmentation accuracy, normalized mutual information (NMI), and adjusted rand index (ARI) following video scene segmentation research (Mahon & Lukasiewicz, 2024). For evaluating LLM-decoded captions, we used the CLIP text score to measure semantic alignment between generated and ground-truth descriptions.

**Implementation Details** Theoretically, any text-to-video diffusion model can be used for video generation based on the decode captions. In this work, ModelScopeT2V (Wang et al., 2023) was

used as our video generator, performing inference with 30 DDIM steps and adopt a 6.0 classifier-free guidance score. The image encoder and text encoder were initialized using CLIP ViT-H/14 from OpenCLIP (Cherti et al., 2023), and Qwen3-0.6B (Yang et al., 2025) served as the LLM decoder. Further detailed hyperparameter configurations are provided in Supplementary Materials A.1.

Table 1: Quantitative comparison of fMRI-to-video reconstruction methods across three datasets, including a real dataset CC2017 and two synthesized datasets (fMRI-WebVid-Syn and CC2017-Syn).

| Dataset | Model | Video-Based | | Frame-Based | | |
| | | Semantic-Level | | Semantic-Level | | Pixel-Level |
| | | 2-way↑ | 50-way↑ | 2-way↑ | 50-way↑ | SSIM↑ |
| --- | --- | --- | --- | --- | --- | --- |
| CC2017 | Kupershmidt | 0.768 | - | 0.769 | - | 0.140 |
| | f-CVGAN | 0.777 | - | 0.721 | - | 0.108 |
| | MindVideo | 0.847 | 0.197 | 0.796 | 0.174 | 0.176 |
| | NeuroClips | 0.834 | 0.220 | **0.806** | 0.203 | **0.390** |
| | GLFA | 0.841 | 0.182 | 0.775 | 0.116 | 0.173 |
| | Mind-Animator | 0.830 | - | 0.805 | - | 0.321 |
| | **ours** | **0.891** | **0.235** | 0.800 | **0.206** | 0.244 |
| CC2017-Syn | MindVideo | 0.813 | 0.164 | 0.780 | 0.107 | 0.107 |
| | GLFA | 0.877 | 0.181 | 0.752 | 0.087 | 0.124 |
| | **ours** | **0.889** | **0.235** | **0.781** | **0.140** | **0.196** |
| fMRI-WebVid-Syn | MindVideo | 0.788 | 0.117 | 0.735 | 0.122 | 0.095 |
| | GLFA | 0.800 | 0.109 | 0.727 | 0.092 | 0.108 |
| | **ours** | **0.819** | **0.122** | **0.803** | **0.138** | **0.129** |

## 4.2 COMPARISON RESULTS

We compare our method against three fMRI-to-video baselines: f-CVGAN(Wang et al., 2022), Kupershmidt (Kupershmidt et al., 2022), MindVideo (Chen et al., 2023), NeuroClips (Gong et al., 2024), and GLFA (Li et al., 2024). Visual comparisons are shown in Figure 3, and quantitative results are presented in Table 1.

According to multi-shot video reconstruction results in Table 1, our method outperforms baselines, particularly in semantic-level metrics. Specifically, on the fMRI-WebVid-Syn dataset, our method achieves a 9.3% improvement in frame-based 2-way classification score compared to the best baseline, while the 50-way classification score shows a substantial 13.1% improvement. Although our method is mainly designed for multi-shot reconstruction, it is also compatible with single-shot data and outperforms baselines in some semantic-level metrics (Table 9).

The visual comparisons in Figure 3 reveal the superior performance of our method in decoding the primary semantics from both single-shot and multi-shot fMRI signals, while baseline methods exhibit significant quality degradation and fail to reconstruct coherent multi-shot sequences. For example, in the fMRI-WebVid-Syn dataset, we precisely reconstruct the complex action "a man holding a gun," whereas GLFA only captures a person, missing the crucial activity. Furthermore, our framework excels at handling scene transitions, accurately depicting the shift from "a woman petting a dog" to "a turtle swimming in the ocean." In contrast, MindVideo and GLFA fail to decode this transitions. These results confirm that our shot-level paradigm enables visually coherent and temporally accurate multi-shot reconstruction.

## 4.3 EXTENSIVE QUALITATIVE RESULTS

In this section, we present additional qualitative results focusing on multi-shot scenarios. A core component of our training approach is a data synthesis strategy that enhances temporal diversity by randomizing scene duration ratios, e.g., (1,3), (2,2), (3,1) for CC2017-Syn dataset. The results in Figure 4 and Figure 5 present reconstruction performance across these varied configurations. It is important to note that the selected frames in the figures are intended to visually represent the relative duration ratios and do not correspond to the total frame count of the final reconstructed videos.

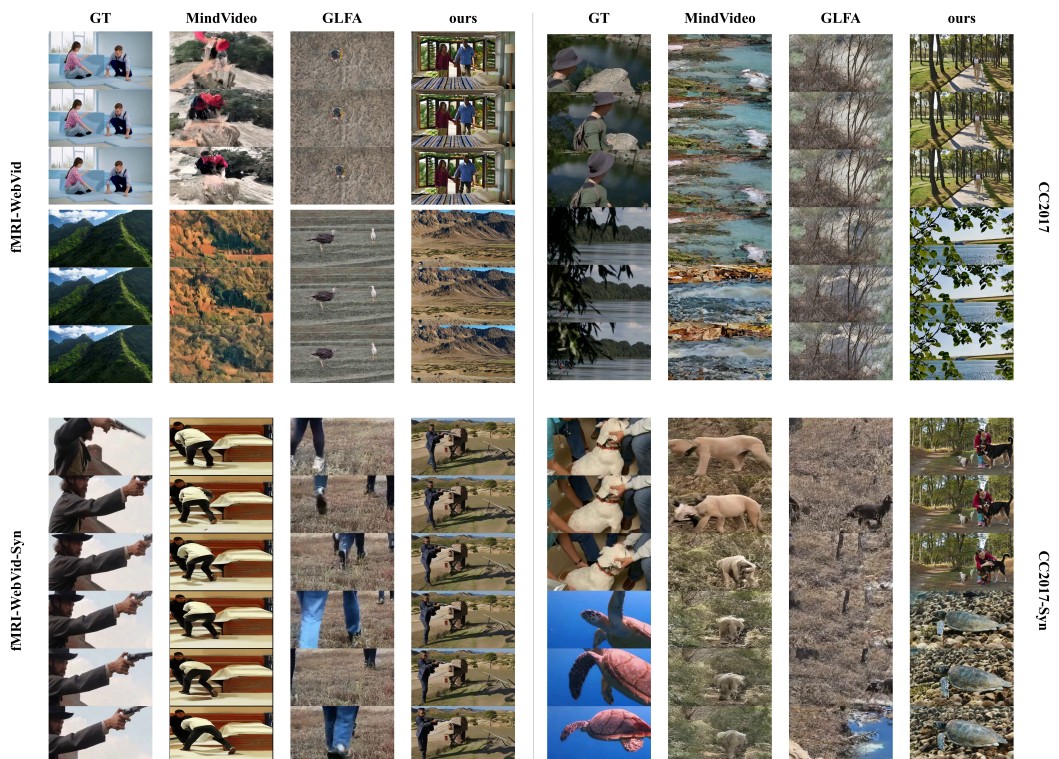

Figure 3: Qualitative comparison of fMRI-to-video reconstruction results on four datasets.

**Robustness to Variable Scene Ratios** As shown in Figure 4 and Figure 5, our method exhibits strong robustness to varying scene durations. By training with diversified scene ratios, our shot boundary predictor effectively overcomes bias towards fixed temporal priors. The model accurately localizes transitions regardless of their temporal position, whether late or early. This confirms that our predictor relies on intrinsic neural pattern changes rather than overfitting to specific timestamps, validating the efficacy of the proposed data augmentation in enhancing temporal generalization. More detailed analysis of data synthesis is shown in Section 4.4.4.

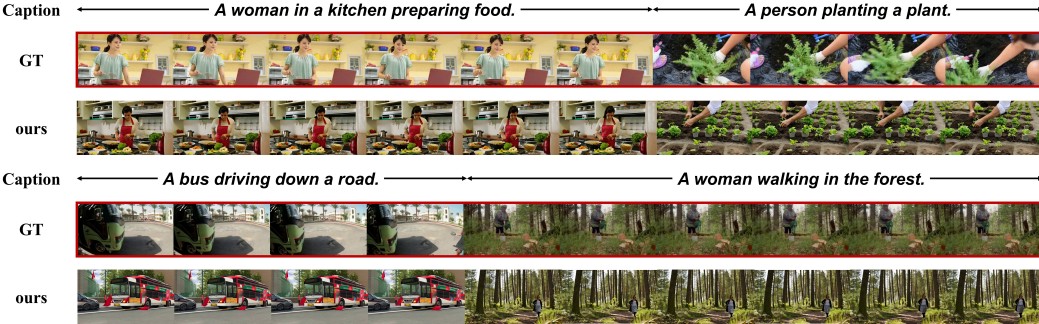

Figure 4: Qualitative results of multi-scene video reconstruction with varying scene ratios on fMRI-WebVid-Syn dataset. Each example displays the text caption (top), the ground truth video frames (middle, red outline), and our reconstruction results (bottom). The black arrows in the caption indicate the duration of each scene.

**Semantic Fidelity Across Complex Transitions** Our method maintains high semantic fidelity even during sharp context shifts, successfully decoding core semantics from noisy fMRI signals. It effectively handles both subtle transitions, such as the "Soldier" to "Horse" sequence where the object

change is correctly identified against a similar background, and drastic transitions with large semantic gaps, exemplified by the shifts from a "Bus" to a "Woman in forest" (Figure 4) or from a "Boat" to a "Basketball game" (Figure 5). Although pixel-level details may vary from the ground truth, the preservation of core semantics across these diverse multi-shot scenarios validates that our paradigm effectively decodes the primary semantics from the continuous, complex fMRI signals.

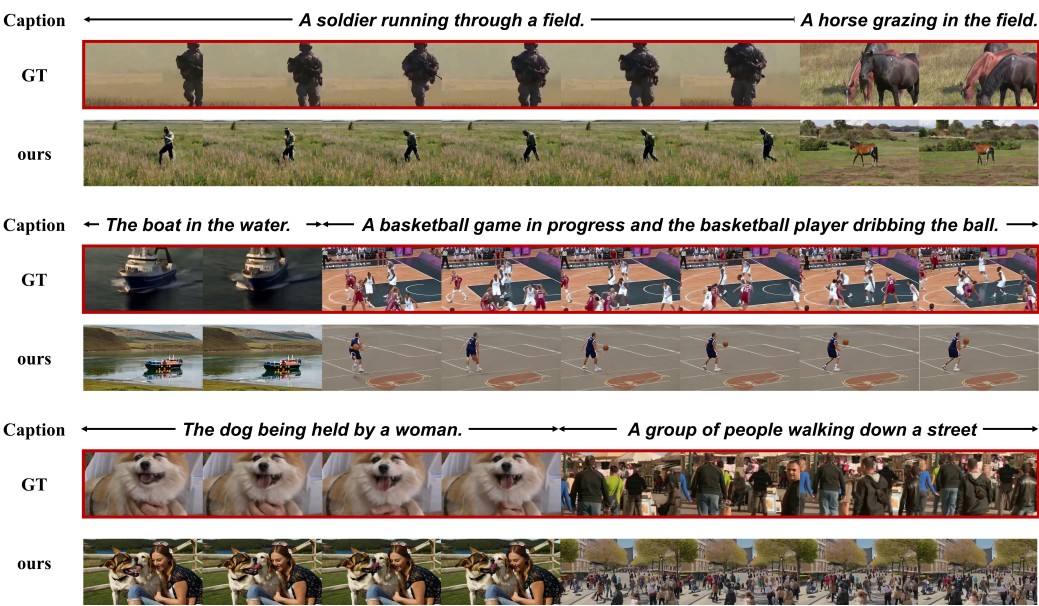

Figure 5: Qualitative results of multi-scene video reconstruction with varying scene ratios on CC2017-Syn dataset.

## 4.4 ABLATION RESULTS

### 4.4.1 IMPACT OF SHOT SEGMENTATION

To evaluate the effectiveness of our proposed shot-specific fMRI segmentation, we conducted comprehensive ablation studies, including improvements by introducing shot boundary predictor and generalizability and robustness analysis.

**Quantitative and Qualitative Improvements**
We compare our full method (w/ $\mathcal{L}_{sbp}$) against a baseline without the shot boundary predictor (w/o $\mathcal{L}_{sbp}$). As shown in Table 2, our shot boundary predictor achieves a segmentation accuracy of 0.685, with 0.683 in ARI and 0.690 in NMI, demonstrating its capability to reliably identify shot transitions from fMRI signals. More importantly, incorporating shot segmentation significantly enhances semantic captioning. Our method improves CLIP similarity by 71.8% compared to the baseline. This confirms that decoding separate keyframe captions per shot yields more semantically precise descriptions than generating a single video-level caption from the mixed fMRI

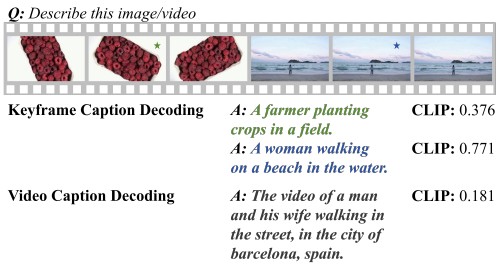

Figure 6: Comparison of decoded captions with and without shot-specific fMRI segmentation.

signal. Qualitative results in Figure 6 further illustrate that captions from the baseline are often vague or biased toward irrelevant content, whereas our shot-aware approach accurately captures the core semantics of each shot.

Table 2: Ablation results of shot-specific fMRI segmentation on fMRI-WebVid-Syn dataset.

| Shot Segmentation | Caption CLIP | Segmentation Metrics | | | Video Reconstruction Metrics | |
|---|---|---|---|---|---|---|
| | | ACC↑ | ARI↑ | NMI↑ | 2-way↑ | 50-way↑ |
| w/o $\mathcal{L}_{sbp}$ | 0.177 | - | - | - | 0.814 | 0.112 |
| w/ $\mathcal{L}_{sbp}$ | 0.304 | 0.685 | 0.683 | 0.690 | 0.819 | 0.122 |

**Generalizability and Robustness Analysis** Beyond our specific architecture, we validated the utility and stability of the SBP module through two additional analyses. First, we integrated the SBP module into other leading methods across two distinct datasets. The consistent performance gains observed in these experiments (Table 3 and Table 7) demonstrate that our shot segmentation strategy is not merely a model-specific optimization but a generalized enhancement module capable of solving the temporal-mixing problem inherent in multi-shot fMRI decoding tasks. V-S 2-way in Table 3 denotes Video-Based Semantic-Level 2-way accuracy, and F-S 2-way denotes Frame-Based Semantic-Level 2-way accuracy. By explicitly modeling temporal boundaries, the SBP allows existing encoders to focus on temporally coherent signal segments rather than noisy, aggregated data. We also introduced temporal noise during inference by shifting predicted boundaries to validate the sensitivity of our model to segmentation errors. The results in Table 8 (A.2) indicate that our model has negligible performance degradation even under high noise settings, suggesting that our decoding modules learn a semantic resilience that tolerates minor temporal misalignments.

Table 3: Ablation results of applying SBP to other fMRI-to-video reconstruction models.

| Dataset | Model | V-S 2-way↑ | V-S 50-way↑ | F-S 2-way↑ | F-S 50-way↑ |
|---|---|---|---|---|---|
| CC2017-Syn | GLFA | 0.844 | 0.151 | 0.703 | 0.060 |
| | GLFA w/ sbp | 0.864 | 0.192 | 0.722 | 0.093 |
| | MindVideo | 0.786 | 0.119 | 0.750 | 0.108 |
| | MindVideo w/ sbp | 0.824 | 0.153 | 0.793 | 0.118 |

### 4.4.2 IMPACT OF LLM DECODING

To evaluate the effectiveness of caption decoding against contrastive alignment for semantic extraction from fMRI, we conduct ablation experiments evaluating CLIP similarity under different training settings. As shown in Table 4, caption decoding improves CLIP similarity by 7.7% over the alignment baseline, demonstrating that decoding text descriptions better reconstruct semantics by mitigating temporal ambiguity. Although caption decoding outperforms the alignment baseline, the reconstruction results are further enhanced by the multi-task framework that combines alignment and decoding. We attribute this to the complementary information provided by different tasks, where alignment task helps preserve structural details while decoding primarily captures semantics. Detailed ablation results for various task combinations are provided in Supplementary Materials A.4, with a more extensive analysis of the decoded captions presented in A.7.

Table 4: Ablation results of evaluating the contribution of LLM-based semantic decoding against contrastive alignment on CC2017 dataset. The segmentation metric reports the average of NMI, ARI, and ACC scores.

| $L_{caption}$ | $L_{align}$ | CLIP↑ | Seg↑ | V-S 2-way↑ | V-S 50-way↑ | F-S 2-way↑ | F-S 50-way↑ |
|---|---|---|---|---|---|---|---|
| - | ✓ | 0.312 | 0.414 | 0.863 | 0.191 | 0.759 | 0.087 |
| ✓ | - | 0.336 | 0.439 | 0.872 | 0.200 | 0.750 | 0.091 |
| ✓ | ✓ | 0.345 | 0.477 | 0.893 | 0.221 | 0.801 | 0.172 |

### 4.4.3 IMPACT OF PROMPT SETTINGS

Our method uses decoded keyframe captions as input prompts for the video generation model. To validate this design choice, we compare three prompt configurations: fMRI-only, text-only, and

dual-modal. The dual-modal approach combines fMRI embeddings and text embeddings of decoded captions with equal weighting. As shown in Table 5, using only decoded keyframe captions as prompts achieves the best results. While the Dual-Modal setting outperforms the fMRI-only baseline, its performance is still lower than the Text-Only setting. We attribute this degradation to the inherent property of fMRI signals, which are known to preserve primary semantics more robustly than high-fidelity visual details (Doerig et al., 2025). Since our LLM-decoded caption already captures the core semantics more accurately than the raw fMRI embedding (as confirmed in Table 4), introducing the lower-fidelity fMRI embeddings for video generation introduces noise and potentially degrades the visual quality, thus reversing the performance gain.

Table 5: Ablation results of different prompt settings for video diffusion model on fMRI-WebVid-Syn dataset.

| Prompt | V-S 2-way↑ | V-S 50-way↑ | F-S 2-way↑ | F-S 50-way↑ |
|---|---|---|---|---|
| fMRI Only | 0.810 | 0.097 | 0.790 | 0.130 |
| Text Only | **0.822** | **0.147** | 0.815 | **0.181** |
| Dual-Modal | 0.809 | 0.108 | **0.821** | 0.171 |

### 4.4.4 GENERALIZATION OF SYNTHETIC DATA

To support large-scale training for multi-shot fMRI-to-video reconstruction, we introduced a novel data synthesis strategy in Section 3.1. Here, we systematically evaluate its effectiveness under three training settings: (1) synthetic, where the model is trained exclusively on synthetic data and evaluated on the real CC2017 dataset; (2) mixed-balanced, which utilizes a balanced mixture of synthetic and real data for training; and (3) mixed-augmented, where the training data is enriched with a higher proportion of synthetic samples. The experimental results in Table 6 indicated that, training solely on synthetic data achieves performance comparable to using real data, demonstrating strong zero-shot transfer and high fidelity of our synthesis pipeline. Moreover, incorporating synthetic data into training, particularly at higher ratios, further improves performance, indicating its effectiveness as a data augmentation strategy that enhances diversity and robustness.

Table 6: Generalization evaluation on synthetic vs. real data configurations. The segmentation metric reports the average of NMI, ARI, and ACC scores.

| Data | CLIP↑ | Seg↑ | V-S 2-way↑ | V-S 50-way↑ | F-S 2-way↑ | F-S 50-way↑ |
|---|---|---|---|---|---|---|
| Real | 0.342 | 0.446 | 0.862 | 0.200 | 0.783 | 0.126 |
| Mixed-Balanced | 0.356 | 0.453 | 0.867 | 0.208 | 0.785 | 0.138 |
| Mixed-Augmented | 0.360 | 0.475 | 0.874 | 0.227 | 0.800 | 0.141 |
| Synthetic | 0.349 | 0.428 | 0.849 | 0.172 | 0.776 | 0.137 |

## 5 LIMITATION AND FUTURE WORK

In this work, we establish a novel shot-level paradigm for multi-shot video reconstruction via shot-specific fMRI segmentation. However, several limitations remain to be addressed in future research, including enhancing the Shot Boundary Predictor to mitigate failure modes and exploring generalizable methods for intra-TR decomposition.

## 6 CONCLUSION

In this work, We propose a divide-and-decode framework for high-fidelity, multi-shot video reconstruction from fMRI. Our approach segments shot-specific fMRI to clearly separate mixed shot-specific signals, and decodes keyframe captions to effectively resolve temporal ambiguity and reduce noise in fMRI signals. Integrating these innovations enables high-quality reconstruction, and we hope this pioneering work inspires future research in multi-shot fMRI decoding.

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

## A SUPPLEMENTARY MATERIALS

### A.1 EXPERIMENTAL DETAILS

**Dataset** CC2017 (Chen et al., 2023) contains three subjects with fMRI frames acquired using a 3T scanner (TR=2s), where each sample includes a 6s video and 3 fMRI scans. fMRI-WebVid (Li et al.,

2024) involves five subjects with fMRI data acquired using a 3T scanner sampled at 1 frame per 0.8s. Stimuli videos (596×336) are sourced from WebVid (Bain et al., 2021), with each sample containing a 4s video and 5 fMRI scans. For original CC2017, we processed 3 fMRI scans to generate 6s videos at 3 FPS. CC2017-Syn used 4 fMRI scans for 6s/6 FPS output. Original fMRI-WebVid processed 5 fMRI scans into 4s/3 FPS videos, while its synthesized counterpart used 5 fMRI scans for 4s/6 FPS reconstruction. All videos were generated at dimensions of 576×320.

**fMRI Preprocessing** Following (Qian et al., 2023), each fMRI scan was projected to 32k_fs_LR brain surface space through anatomical structure and transformed to a 256×256 single-channel image, where only early and higher cortical regions retained values. fMRI data were averaged across multiple runs for the same video in both datasets.

**Hyperparameter Settings** The fMRI encoder in our work have stacks of transformer blocks with depth of 24, feature dimensions of 1024, and 16 multi-heads in self-attention layers. We initialize them with pretrained weights from (Li et al., 2024). For LLM decoding, the features from fMRI encoder input into a mapper layer with two fully-connected layer, to compress the feature dimension to $77 \times 1024$. During training, we adopt Adam optimizer with $\beta_1 = 0.9$, $\beta_2 = 0.999$. The initial learning rate is set to 0.0001.

## A.2 GENERALIZABILITY AND ROBUSTNESS OF SHOT SEGMENTATION

To evaluate the effectiveness of our proposed shot-specific fMRI segmentation, we conducted comprehensive ablation studies, including improvements by introducing shot boundary predictor and generalizability and robustness analysis. The corresponding tables presenting these additional results are provided in this section to complement the analysis in Section 4.4.1.

Table 7: Ablation results of applying shot boundary predictor to other fMRI-video reconstruction models.

| Dataset | Model | V-S 2-way↑ | V-S 50-way↑ | F-S 2-way↑ | F-S 50-way↑ |
|---|---|---|---|---|---|
| fMRI-WebVid-Syn | GLFA | 0.825 | 0.110 | 0.699 | 0.090 |
| | GLFA w/ sbp | 0.825 | 0.128 | 0.744 | 0.138 |
| | MindVideo | 0.792 | 0.112 | 0.760 | 0.069 |
| | MindVideo w/ sbp | 0.816 | 0.136 | 0.762 | 0.118 |

Given that our LLM decoding relies on segmentation, we assessed the model's sensitivity to segmentation errors. We introduced temporal noise during inference by shifting predicted boundaries by 1 TR (low noise) and 2 TRs (high noise). The results in Table 8 indicate negligible performance degradation, suggesting that our decoding modules learn a semantic resilience that tolerates minor temporal misalignments. This robustness implies that the model captures the core semantic gist of a shot even when the precise onset and offset are slightly perturbed, making it highly practical for real-world applications where perfect segmentation is challenging.

Table 8: Robustness evaluation under simulated noisy shot segmentation on the CC2017 dataset.

| Noise | CLIP↑ | V-S 2-way↑ | V-S 50-way↑ | F-S 2-way↑ | F-S 50-way↑ |
|---|---|---|---|---|---|
| High | 0.332 | 0.877 | 0.202 | 0.796 | 0.136 |
| Low | 0.345 | 0.881 | 0.240 | 0.796 | 0.128 |
| wo/ noise | 0.356 | 0.893 | 0.221 | 0.801 | 0.172 |

## A.3 COMPARISON RESULTS ON SINGLE-SHOT DATASET

Our method is designed for multi-shot reconstruction but is also compatible with single-shot data. The comparative results on fMRI-WebVid dataset (Table 9) show that our approach achieves superior performance on most metrics compared to the second-best model, further indicating the effectiveness and general applicability of LLM-based caption decoding in single-shot dataset.

Table 9: Quantitative comparison of fMRI-to-video reconstruction methods on fMRI-WebVid dataset.

| Dataset | Model | Video-Based | | Frame-Based | | |
| | | Semantic-Level | | Semantic-Level | | Pixel-Level |
| | | 2-way↑ | 50-way↑ | 2-way↑ | 50-way↑ | SSIM↑ |
|---|---|---|---|---|---|---|
| | MindVideo | 0.736 | 0.075 | 0.760 | 0.109 | 0.097 |
| fMRI-WebVid | GLFA | **0.806** | 0.118 | 0.809 | 0.173 | **0.188** |
| | **ours** | 0.790 | **0.135** | **0.817** | **0.183** | 0.145 |

## A.4 ABLATION RESULTS OF LOSS FUNCTION

Table 10: Ablation results on semantics extraction methods on fMRI-WebVid-Syn dataset.

| Loss Function | | | Metric |
| $\mathcal{L}_{caption}$ | $\mathcal{L}_{align}$ | $\mathcal{L}_{mse}$ | CLIP↑ |
|---|---|---|---|
| - | ✓ | - | 0.283 |
| - | ✓ | ✓ | 0.280 |
| ✓ | - | - | 0.302 |
| ✓ | ✓ | - | 0.313 |
| ✓ | - | ✓ | 0.300 |
| ✓ | ✓ | ✓ | **0.336** |

While Section 4.4.2 ablates the contribution of LLM-based decoding against contrastive alignment, our full implementation employs a combination of four loss functions. We therefore provide a complete ablation study here for a comprehensive analysis. As shown in Table 10, the superiority of the captioning loss $L_{caption}$ over $L_{align}$ is consistent, regardless of the introduction of the MSE loss $L_{mse}$. Furthermore, although the U-Net remains frozen, incorporating $L_{mse}$ during training is shown to enhance the semantic fidelity of the decoded content, underscoring the utility of noise prediction as an effective training guide.

To further clarify the individual contributions of shot boundary predictor and LLM-based decoding, we have included complete ablation results in Table 11. Our experiments show that introducing shot segmentation alone leads to an 82.5% improvement in the caption CLIP score. In comparison, the additional use of LLM-based decoding further improves performance by 10.6%.

Table 11: Ablation results of individual improvments of shot segmentation and LLM-based decoding on CC2017 dataset.

| $L_{sbp}$ | $L_{caption}$ | CLIP↑ | V-S 2-way↑ | V-S 50-way↑ | F-S 2-way↑ | F-S 50-way↑ |
|---|---|---|---|---|---|---|
| - | ✓ | 0.171 | 0.822 | 0.173 | 0.750 | 0.094 |
| ✓ | - | 0.312 | 0.863 | 0.191 | 0.759 | 0.087 |
| ✓ | ✓ | 0.345 | 0.893 | 0.221 | 0.801 | 0.172 |

## A.5 ANALYSIS OF CROSS-SUBJECT GENERALIZATION

Our fMRI encoder follows the architecture of (Li et al., 2024), operating in a unified brain space to enable cross-subject decoding. To evaluate cross-subject generalization, we performed per-subject inference. As shown in Table 12 and Table 13, the results exhibit high consistency across different subjects on all datasets, demonstrating the strong robustness of our approach.

## A.6 ANALYSIS OF FAILURE CASES

While our multi-shot reconstruction paradigm demonstrates high performance in general cases, we acknowledge and analyze three primary failure cases, as illustrated in Figure 7.

Table 12: Per-subject inference results on CC2017 dataset, where the segmentation metric is the average of NMI score, ARI score, and ACC.

| Subject | CLIP ↑ | Seg ↑ | V-S 2-way ↑ | V-S 50-way ↑ | F-S 2-way ↑ | F-S 50-way ↑ |
|---------|--------|-------|-------------|--------------|-------------|--------------|
| Subject 1 | 0.345 | 0.477 | 0.893 | 0.221 | 0.801 | 0.172 |
| Subject 2 | 0.317 | 0.438 | 0.853 | 0.207 | 0.761 | 0.124 |
| Subject 3 | 0.341 | 0.440 | 0.874 | 0.192 | 0.771 | 0.128 |

Table 13: Per-subject inference results on fMRI-WebVid-Syn dataset, where the segmentation metric is the average of NMI score, ARI score, and ACC.

| Subject | CLIP ↑ | Seg ↑ | V-S 2-way ↑ | V-S 50-way ↑ | F-S 2-way ↑ | F-S 50-way ↑ |
|---------|--------|-------|-------------|--------------|-------------|--------------|
| Subject 1 | 0.295 | 0.484 | 0.798 | 0.142 | 0.812 | 0.169 |
| Subject 2 | 0.301 | 0.503 | 0.825 | 0.137 | 0.813 | 0.145 |
| Subject 3 | 0.316 | 0.518 | 0.837 | 0.138 | 0.817 | 0.136 |
| Subject 4 | 0.308 | 0.517 | 0.829 | 0.173 | 0.811 | 0.150 |
| Subject 5 | 0.303 | 0.506 | 0.819 | 0.141 | 0.819 | 0.155 |

**False Negatives Due to Low Semantic Disparity (Scene Mixing)** The result in Figure 7(a), represents a false negative in boundary prediction. When the two adjacent scenes exhibit high semantic overlap, the Shot Boundary Predictor may fail to detect the temporal transition. In the example of "Cockatiel on a perch" transitioning to "A tree with a bird," the low semantic disparity between the concepts (both involving nature, trees, and birds) results in the model perceiving a continuous signal. Consequently, the decoder blends the features of both segments, reconstructing a single, merged scene that attempts to incorporate elements from both original concepts (e.g., a bird in a visually complex tree background). This indicates a limitation in distinguishing subtle temporal changes when the visual concepts are semantically adjacent in the feature space.

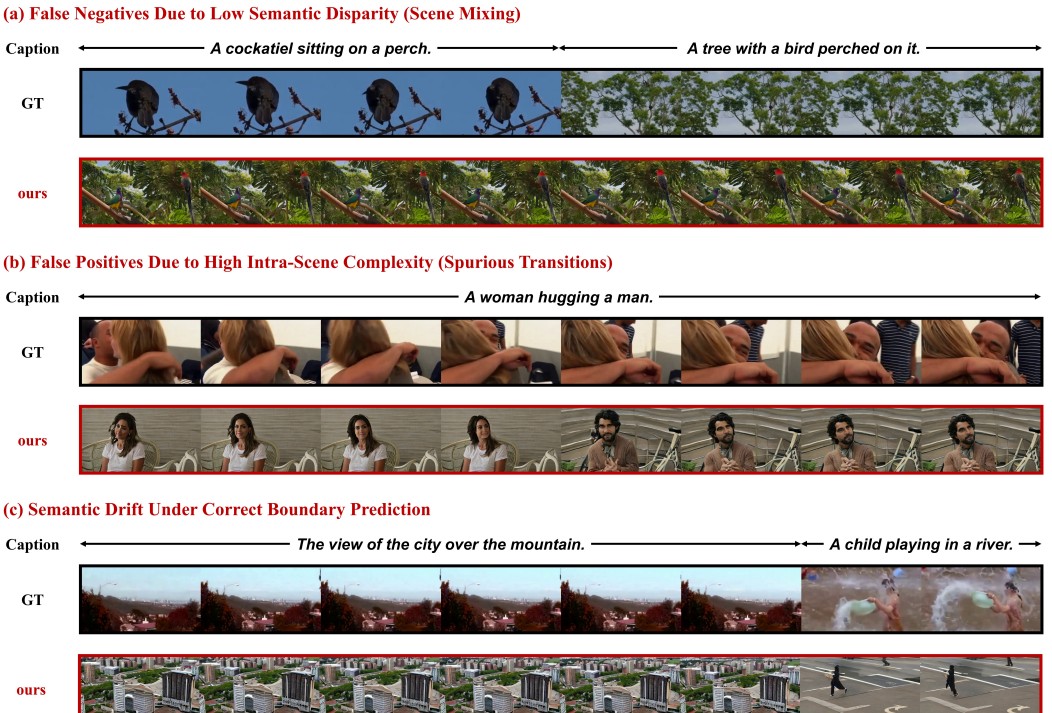

Figure 7: Qualitative examples of fMRI-to-video reconstruction results on CC2017 datasets.

**False Positives Due to High Intra-Scene Complexity (Spurious Transitions)** Conversely, Figure 7(b) illustrates a false positive error, where the model mistakenly predicts a boundary within a single, continuous shot. In the example of "A woman hugging a man," the continuous scene contains high visual entropy—significant subject movement, changing light patterns, and complex object occlusion. These rapid, complex visual changes lead the Shot Boundary Predictor to interpret the shifts in fMRI signal dynamics as an abrupt cut, resulting in the false segmentation of the single action into two distinct, static scenes (a woman alone and a man alone). This highlights a trade-off: high sensitivity to rapid neural pattern changes can lead to desirable boundary detection but risks over-segmentation in visually turbulent single shots.

**Semantic Drift Under Correct Boundary Prediction** The final failure mode, Figure 7(c), demonstrates the difficulty in decoding accurate semantics even when the temporal localization is correct. In this example, the model successfully recognizes the scene boundary (a true positive in segmentation). However, the reconstructed content is highly inaccurate. The abstract concept "The view of the city over the mountain" yields a detailed, yet incorrect, modern apartment block, while the subsequent scene "A child playing in a river" is rendered as generic dark pavement and water. This semantic drift is often attributable to the low intrinsic quality or high ambiguity of the original scene's fMRI signature (e.g., "city view" is an over-generalized concept). A weak mapping in the decoding phase for less frequently represented visual concepts in the training data. This underscores that while our architecture effectively solves the multi-shot temporal challenge, the final reconstruction fidelity remains constrained by the quality and specificity of the available fMRI-caption pairs. Future work necessitates the collection of higher-quality, less ambiguous fMRI data to mitigate this decoding error.

## A.7 ANALYSIS OF DECODED CAPTIONS

**Quality Analysis** To analyze the quality of the LLM-generated captions, we manually inspected a randomly selected subset of decoded captions (N=204). We observed that the model consistently recovers high-level semantic categories and actions (e.g., "a man walking in a park", "a turtle swimming in an ocean") but often omits or mis-specifies low-level attributes such as exact colors, small background objects, or fine-grained object subtypes. Following (Wang et al., 2025), we computed these NLP metrics to quantify this observation (as shown in Table 14): relatively high BLEU@1 and SPICE scores, indicating strong capture of semantic content; weaker CIDEr scores, which are more sensitive to extract n-gram matches and fine-grained details.

Importantly, the tendency to produce semantically appropriate but syntactically generic captions is not merely a limitation of an unconstrained LLM. It is also consistent with established findings that brain activity reflects abstract semantic information rather than low-level visual details. Therefore, the fact that our decoded captions capture the correct semantic gist while missing fine-grained attributes is consistent with the information recorded in brain regions, suggesting that the decoded captions from LLM are based on the fMRI signals rather than solely on the language priors of LLM.

Table 14: Linguistic metrics analysis of decoded captions.

| Metric | CC2017 | CC2017-Syn | WebVid | WebVid-Syn |
|---|---|---|---|---|
| BLEU@1 | 22.3 | 26.4 | 25.9 | 24.8 |
| ROUGE-L | 25.6 | 25.8 | 25.9 | 27.3 |
| CIDEr | 16.2 | 17.0 | 16.5 | 20.9 |
| SPICE | 14.6 | 14.5 | 15.5 | 16.2 |

**Hallucination Analysis** To further assess potential hallucinations in the LLM-generated captions, we conducted a human evaluation categorized into three types, including object, attribute, and scene or activity hallucination.

The evaluation results highlight that object hallucination is the most prominent issue. Specifically, the model frequently confused semantically similar but distinct entities. For instance, "a turtle swimming in the water" was incorrectly described as "a fish swimming in the water," and "a tiger in the snow" was erroneously captioned as "a tiger in the snow with a dog." This suggests a potential

weakness in the LLM's fine-grained visual discrimination, possibly stemming from the ambiguity inherent in the fMRI-derived features or the LLM's reliance on highly frequent co-occurrence patterns learned during pre-training, where fish might be a more common aquatic object than turtle. Scene or activity hallucination ranks second, accounting for 23% of the identified errors. Typical examples include substituting fine-grained concepts with broader categories, such as describing "jogging" as "walking," "beach" as "field," and "walking down the street" as "walking in a park." This trend indicates that the model excels at capturing high-level semantic concepts but struggles with specific contextual details (e.g., the speed of motion, the exact type of location). These coarse-grained descriptions may limit the fidelity of subsequent video reconstruction stages.

**Social Bias Assessment** Regarding social biases, no obvious or harmful stereotypical representations were observed in the sampled captions. However, we did note a recurring pattern in attire description: men were frequently depicted wearing suits, while women were often described as wearing dresses. While not overtly harmful, this pattern suggests the LLM has absorbed and reproduced imbalanced co-occurrence statistics prevalent in its training data concerning gender and professional attire.

## A.8 LLM USAGE STATEMENT

Large Language Models (LLMs) were used to aid in the writing and polishing of the manuscript. Specifically, we used an LLM to assist in refining the language, improving readability, and ensuring clarity in various sections of the paper. The model helped with tasks such as sentence rephrasing, grammar checking, and enhancing the overall flow of the text. It is important to note that the LLM was not involved in the research methodology, or experimental design. All research concepts, ideas, and analyses were developed and conducted by the authors. The contributions of the LLM were solely focused on improving the linguistic quality of the paper, with no involvement in the scientific content or data analysis. The authors take full responsibility for the content of the manuscript. including any text generated or polished by the LLM. We have ensured that the LLM-generated text adheres to ethical guidelines and does not contribute to plagiarism or scientific misconduct.

